# Temporal Analysis Reveals the Transient Differential Expression of Transcription Factors That Underlie the Trans-Differentiation of Human Monocytes to Macrophages

**DOI:** 10.3390/ijms232415830

**Published:** 2022-12-13

**Authors:** Weihang Deng, Min Chen, Ying Tang, Le Zhang, Zeqian Xu, Xinhui Li, Daniel M. Czajkowsky, Zhifeng Shao

**Affiliations:** State Key Laboratory for Oncogenes and Bio-ID Center, School of Biomedical Engineering, Shanghai Jiao Tong University, Shanghai 200240, China

**Keywords:** mononuclear phagocyte system, trans-differentiation, macrophage, time-series analysis

## Abstract

The activation of monocytes and their trans-differentiation into macrophages are critical processes of the immune response. Prior work has characterized the differences in the expression between monocytes and macrophages, but the transitional process between these cells is poorly detailed. Here, we analyzed the temporal changes of the transcriptome during trans-differentiation of primary human monocytes into M0 macrophages. We find changes with many transcription factors throughout the process, the vast majority of which exhibit a maximally different expression at the intermediate stages. A few factors, including AP-1, were previously known to play a role in immunological transitions, but most were not. Thus, these findings indicate that this trans-differentiation requires the dynamic expression of many transcription factors not previously discussed in immunology, and provide a foundation for the delineation of the molecular mechanisms associated with healthy or pathological responses that involve this transition.

## 1. Introduction

Monocytes and macrophages are key components of the immune system that play vital roles in maintaining a healthy condition but also in directly affecting a number of diseases, including those associated with chronic inflammation and sepsis [1,2]. In one of the primary pathways by which these cells function, circulating monocytes infiltrate into tissues, become activated, and then trans-differentiate into macrophages, which then function as a complement to the tissue-resident macrophages, contributing to phagocytosis, antigen presentation, the secretion of cytokines, and other functions [3,4]. Yet, while of crucial significance to human health, this monocyte-to-macrophage trans-differentiation process remains poorly detailed, which has hindered our understanding of the critical molecular mechanisms underlying the innate immune response more generally, as well as the development of therapies that could mitigate this process during the pathological responses [1,5].

Most studies to date of the human monocyte-to-macrophage trans-differentiation process have focused on in vitro investigations using immortal monocytic cell lines (such as THP-1 and U937 [6,7,8]) or primary blood monocytes [9,10,11]. However, it is now well appreciated that there are significant differences between transformed, cultured cells and their primary counterparts, and so studies of the latter are much preferred to more correctly understand the processes in vivo. In a well-established in vitro approach that mirrors the process in vivo [3,12], purified primary monocytes are allowed to first adhere to the surface of polystyrene cell culture flasks, which triggers their transition to a surface-activated state that is necessary to enable the subsequent monocytes-to-macrophages trans-differentiation process, as well as to promote cell survival [13,14]. This step resembles the activation of monocytes in vivo, which occurs during their transition from a non-adherent state to the adherent state [15,16]. The subsequent addition of the maturation factors, such as normal human serum (NHS) then initiates the trans-differentiation of the activated monocytes into resting naïve macrophages (M0) in seven days [17,18,19].

With approaches such as this, many differences in the transcriptomes between the initial monocytes and the final, trans-differentiated macrophages have been described [6,7,20,21]. A common focus of these studies is the transcription factors (TFs), owing to their fundamental role in driving many differentiation processes [22]. Indeed, this prior work has identified a number of TFs that have been shown to play a role in this trans-differentiation, including PU.1, C/EBPs and MAFB [23,24,25]. However, while informative of the important differences between the initial and final cell types along this pathway, the critical transitional process between these cells has not yet been carefully examined. There is thus, essentially, no knowledge of any TFs that might be only transiently expressed during this process. Indeed, differentiation processes that involve the transformation of a pluripotent cell (such as those occurring during early embryogenesis [26,27] and organ development [28]) are well known to occur via the dynamic expression of “waves” of TFs throughout the process. Thus, it is critical to determine whether this trans-differentiation, which occurs by the transformation of one somatic immune cell type to another without cell division [29], also involves such complex, dynamic changes in the TF expression.

To this end, we examined the changes to the transcriptome during the surface-activation of human primary monocytes and their trans-differentiation into naïve M0 macrophages, following the addition of NHS with hour-level temporal resolution. Unexpectedly, based on the transcriptomic data alone, the differentiation process is essentially complete in the first day. Among the differentially transcribed genes during this process, are more than 400 TFs, roughly half showing the enhanced expression and half showing a reduced expression. Most of these TFs achieve their maximal difference in expression at an intermediate stage during this transition, consistent with “waves” of transiently differentially expressed TFs underlying this trans-differentiation, similar to the embryonic differentiation processes [27]. Among these are members of the well-known AP-1 family of TFs, most of which are up-regulated immediately with surface-activation but are then down-regulated almost immediately thereafter. However, the majority of these TFs were not previously implicated in this trans-differentiation, or in any immunological process, thereby significantly expanding the TF repertoire that are involved in this transition and in immunology. Hence, overall, we show that the trans-differentiation of primary monocytes to M0 macrophages is associated with an unexpectedly complex, dynamic pattern of transiently differentially expressed TFs, most of them new to immunology, providing a foundation for the delineation of the molecular mechanisms associated with healthy or the pathological responses that involve this transition.

## 2. Results

### 2.1. Temporal Profiling of the Transcriptomic Changes during the Trans-Differentiation of Primary Monocytes to M0 Macrophages In Vitro

To delineate the temporal evolution of the transcriptome during the trans-differentiation of primary monocytes to macrophages, we adapted an already established in vitro protocol of trans-differentiation and examined the transcriptome at many time-points during this transition [19,30]. In short, human primary monocytes were isolated to a high purity (Appendix A), added to a serum-free medium within the polystyrene flasks to allow for the surface adherence, and then induced into M0 macrophages with the addition of NHS. As documented previously [4,31], the phase-contrast and immunofluorescence microscopy confirmed the typical features of the M0 macrophages after seven days of incubation (Appendix A). In particular, after this time, the cells showed a high CD68 expression and a low CD14 expression, and the shape of the nuclei changed from a U-shape (monocytic) to a rounded (macrophagic) shape. In addition, the nuclear volume increased and the nuclei became more spherical, also as expected (Appendix A).

To examine the temporal evolution in the transcriptome during this process, we performed deep RNA sequencing with the purified monocytes (mono), the surface-activated monocytes before the addition of NHS (0 h), and cells at nine time points following the NHS addition (2 h, 4 h, 8 h, 12 h, 1-day, 2-days, 3-days, 4-days and 7-days) (Figure 1A). For each sample, 30.8 million high quality reads, on average, were obtained (Appendix A). We verified the quality of our data by comparing with the previously published results [32] of purified monocytes and the differentiated M0 macrophages (7-days), finding an excellent agreement (Appendix A). Comparing the transcriptomes of just the purified monocytes and macrophages, we identified 2747 differentially expressed mRNAs (*p.* adjust < 0.01, |log2(FC)| > 2). The GO analysis revealed that these differentially expressed genes (DEGs) were enriched for several immunology-relevant terms such as the inflammatory response and cell-adhesion regulation (Appendix A), as expected.

### 2.2. PCA Analysis Indicates That the Transcriptional Changes during This Transition Are Largely Completed in One Day

We employed the principal component analysis (PCA) to examine the relationship between the transcriptomes obtained at all measured time points (Figure 1B and Appendix A). This analysis revealed two unexpected details. First, the greatest difference between the subsequently measured time points in our dataset occurred between the purified monocytes (mono) and the surface-activated monocytes (0 h). While, as expected, the changes that occurred subsequently were, in toto, more significant than during this initial step, this finding suggests that the initial step involving the surface-binding induced activation is indeed associated with a dramatic change to the transcriptome. Second, the transcriptome after only one day in this process is highly similar to that observed in the fully differentiated M0 macrophage. This result is further confirmed by an examination of the pairwise correlations of the transcriptomic data from all samples, which shows an exceptionally high degree of correlation (Spearman correlation coefficients: 0.94 to 0.98) in the transcriptomes between the samples from day one to day seven (Appendix A). Hence, these results suggest that, at least at the level of the transcriptome, most of the changes that occur during this transition are completed after only about one day, long before the changes in the cell morphology were complete (seven days). While it is obvious that some time is needed to adjust the proteome and cell morphology, according to the transcriptomic changes, it was unexpected that, in fact, the bulk of the time (~6 of the 7 days) that is conventionally thought to be needed for the full differentiation to the macrophage phenotype is actually primarily associated with the events subsequent to the transcriptomic alterations. We note that these observations were confirmed by the PCA analysis of just a subset (1042) of the aforementioned 2747 DEGs that exhibited a high expression level (TPM > 10 in at least one sample of the monocytes or macrophages) and were not housekeeping genes, as these transcripts might contribute most significantly to the changes in phenotype between these cells (Figure 1C).

### 2.3. Hierarchical Clustering Shows “Wave-Like” Changes in the Transcription of TFs during the Trans-Differentiation

While we expect that many pathways undergo changes during this transition, since we were primarily interested in the underlying molecular mechanisms driving this process, we focused here on a more detailed examination of the TFs, owing to their dominant role in most differentiation processes. The initial inspection of our data showed that some TFs that were previously described as playing a role in the monocyte-to-macrophage trans-differentiation (such as PU.1 and MAFB), remain at a high level of expression throughout the process (Appendix A). However, we also noticed that many other TFs underwent significant changes in expression during this transition. Based on an established database of the annotated TFs [33], we identified 220 differentially expressed TFs (DE TFs) that were significantly up-regulated and 227 DE TFs that were significantly down-regulated (maximal TPM > 10, |log2(FC)| > 1.5, *p*. adjust < 0.01) during the entire measured time period (Appendix A). We note that hundreds of DE TFs have been previously observed during specific stages of the early embryonic differentiation processes [26,27], indicating that, although this trans-differentiation only involves a transition between two immune cell types, it appears to be as complex a process as the differentiation of the pluripotent stem cells into somatic cell types. Of these DE TFs, the vast majority (372) showed a maximal change in the expression at an intermediate stage of this process (that is, at a time period between 0 h and three days). Thus, the activation of the monocytes and their trans-differentiation to M0 macrophages is associated with the transient differential expression of many TFs, most of which essentially return to their levels in the purified monocytes by the end of the process.

To examine this issue further, we used hierarchical clustering, based on the expression dynamics of the DE TFs (see Materials and Methods). We found that the up-regulated DE TFs formed seven groups and the down-regulated DE TFs formed five groups (Figure 2A,C, Appendix A). Each group exhibited a maximal change in the expression at a different time period, reminiscent of the “waves” of the DE TFs observed in the differentiation processes during early embryogenesis and organ development [27,34]. Of the up-regulated DE TFs, group 1 harbored the largest number of TFs (60) (Figure 2B). This group was significantly up-regulated at 0 h (that is, following the surface-absorption triggered cell activation), but then, following the addition of the NHS, were rapidly down-regulated to their expression levels in the monocytes prior to this activation (Figure 2B). Thus, this group appears to be a unique set of TFs that is associated specifically with the initial step of the transition that includes the surface-activation of the monocytes.

Inspection of the two types of groups also revealed some groups that appeared to change over exactly the same time periods but in precisely the opposite directions (Figure 2A,C). In particular, the expression of the up-regulated group 2 was enhanced from 0 h to 4 h, peaking at 2 h, and then reduced to the initial levels thereafter, while that of the down-regulated group 2 was reduced from 0 h to 4 h, mostly becoming minimal at 2 h, and then increased to the initial levels thereafter. A similar, diametrically opposite temporal profile was also observed between the up-regulated groups 3 and 4 and the down-regulated group 3. Thus, this observation indicates that there may be a coordination between the specific subsets of the up-regulated and down-regulated DE TFs during this trans-differentiation.

We note that we confirmed the validity of these changes in transcription of the TFs by the qRT-PCR, examining four select TFs (OLIG2, EGR4, HIC1 and DLX2) from the up-regulated group 1 at four different time points (mono, 0 h, 8 h and 3-days) (Figure 2D).

### 2.4. Bioinformatics Analysis Identifies the Potential Key Regulator TFs of This Transition

TFs often function via the protein-protein interactions (PPIs) [35]. Further, in many transitions involving significant transcriptional changes, those TFs that are involved in many PPIs have been found to be key drivers of the transition [25,36]. Therefore, we used STRING (v 11.5,European Molecular Biology Laboratory, Heidelberg, Germany) to identify the functionally-associated PPI networks, based on the experimentally validated PPIs, within each individual group. We found that the up-regulated groups 1 and 2 (that peak in expression at 0 h and at 2 h, respectively) exhibited extensive PPI networks (Figure 3A,B). As these are the two groups whose expression is maximal during the surface activation of the monocytes and immediately after the addition of the NHS, respectively, it is possible that these two key initiating events of the trans-differentiation are mostly triggered by the coherently interacting TFs that then dictate the subsequent gene up-regulation.

In the up-regulated group 1, we identified 11 TFs with a high number of putative PPIs (SRF, FOSL2, DDIT3, EGR3, FOSB, RARA, FOSL1, NFE2L2, ATF4, NR4A1, and EGR4) (Figure 3A,C), while in the up-regulated group 2, there was only one, CEBPB (Figure 3B, see Methods). Among these, 11 genes in the up-regulated group 1, four are members of the AP-1 family (FOSB, FOSL1 FOSL2 and ATF4), suggesting that AP-1 TFs play an important role in this process.

We also note that CEBPB, the TF involved in many PPIs in the up-regulated group 2 (Figure 3B), has previously been found to play an important role in macrophage functioning, including specialization [23]. Hence, as this group 2 is associated with the increased expression that follows the addition of the NHS, it may be that this TF plays a dominant role in the initiation of the trans-differentiation to the macrophages, as well as the differentiation of the (final) macrophages.

## 3. Discussion

The transformation of human monocytes to other immune cells plays a fundamental role in innate immunity, but our understanding of the transitional processes that underlies these transitions remains poor. Previous work on the TFs that are involved in this transition in primary cells has focused only on a few genes at a few time points [32,37,38,39]. Here, we provide the first detailed temporal dissection of the transcriptomic changes during the trans-differentiation of primary monocytes into M0 macrophages, particularly focusing on the changes in expression of the TFs.

There were a number of unexpected observations. First, we found that the transcriptomic changes have largely been completed in the first day, far earlier than the morphological changes that have become apparent. Thus, the majority of the time that is typically associated with this transition (six of the seven days) is apparently associated with the reconfiguration of the proteome, their interactions, and other post-transcriptional changes that ultimately produce the final macrophage phenotype.

Second, we found that this entire process is associated with the transient differential expression of nearly 400 TFs, which is similar to the extensive waves of the transiently differentially expressed TFs observed in more complex differentiation processes, such as embryogenesis [26,27]. Thus, while the monocyte-to-macrophage trans-differentiation is only associated with the transition between two somatic immune cells, the degree of changes needed to effect this transformation may be as complicated as those involved in the differentiation of pluripotent cells to specialized somatic cells.

Among these transiently DE TFs, we found that there are a few that have been previously described to play a role in this trans-differentiation or in the mediating macrophage immune responses. These include the NFKB family members (NFKB-1, NFKB-2 and RELB) and CEBPB, which we found to be up-regulated DE TFs during the early stages of this process. Although these TFs were not previously implicated in this transition per se, they were previously shown to be important regulators of the macrophage polarization and the acute inflammatory activation of macrophages [40,41]. In addition, IRF4 and IRF5, both down-regulated DE TFs, have been described to play a role in the differentiation of mature small peritoneal macrophages and macrophage polarization, respectively [22,40].

Moreover, we also found that the members of the well-studied AP-1 family of TFs (namely, FOSL1, FOSB and MAF) were among the up-regulated DE TFs in this process. Previous work has indeed implicated AP-1 TFs in developmental process of many hematopoietic cell lineages, including this monocyte trans-differentiation [42]. In particular, AP-1 was found to be significantly up-regulated in the THP-1 cell line following the addition of PMA (comparing just the initial THP-1 cells with the PMA-induced macrophagic cells) [7]. However, this result contrasted with measurements obtained with the primary human monocytes differentiated into macrophages (by the addition of GM-CSF), which showed that AP-1 was down-regulated, following this transition (comparing just the initial monocytes and the differentiated macrophages) [11]. Here we show that most AP-1 members are indeed significantly up-regulated, but only transiently, with the expression reducing to lower levels following the addition of NHS. Thus, our results are consistent with the previous results obtained with the primary cells, as the levels of these TFs in the macrophage cells are lower than in the monocytes (Appendix A). Yet, our results also suggest that AP-1 indeed plays a critical role in this transition, only increasing the expression transiently, namely during the first step of this process.

However, apart from these few examples of TFs known to play a role in this process or in macrophages, there were in fact many more transiently DE TFs that were not previously reported to function in this trans-differentiation or even within the immune cells. For example, the up-regulated DE TFs group 1 included OLIG2, which was only previously identified as expressed in the central nervous system, specifically in newly differentiated olfactory sensory neurons [43]. In addition, CREM in the up-regulated DE TF group 2, is known to be highly expressed in post-meiotic germ cells where it plays a major role in a transcriptional cascade in human spermatogenesis and spermatid maturation [44]. Further, the down-regulated group 3 DE TF includes PATZ1, thought to be only expressed in the brain, is an essential TF for the maintenance of neural stem cells [45], and the down-regulated group 2 DE TF, CC2D1A, was previously described to play important functions in the early postnatal brain development and the maturation of central synapses [46]. Hence, it is possible that these TFs have been co-opted to play roles in the monocyte trans-differentiation. Likewise, it is also probable that many of the transiently DE TFs described here will be subsequently found to play a role in other differentiation systems as well.

Third, we found that the initial step of this process that includes the surface-activation of the monocytes, is associated with a rapid significant change in the transcriptome, including the up-regulation of many transiently DE TFs. While it is possible that some of the changes observed during this step are owing to the change in media conditions, previous work has shown that the transfer of primary monocytes to media without NHS (without surface binding) does not induce significant changes in the monocytes [47,48]. This is consistent with our speculation that such changes are owing to surface adsorption. Indeed, changes to monocytes consequent to surface binding have also been previously observed [49,50,51]. Of particular note, earlier studies noted an activation of AP-1 upon adherence [49,50]. In our work, we also found that most AP-1 members are up-regulated during the initial step, consistent with these earlier findings, although to our knowledge, the striking subsequent downregulation that we observed was not described before.

Finally, we found that AP-1 TFs may in fact act as important regulators not only of this acute cell activation but also in the early stages following the NHS addition. Beyond the identification of their possible involvement in multiple PPIs (Figure 3a), we also found that the promoter regions of the up-regulated group 1 DE TFs also exhibit an enrichment of the AP-1 transcription factor binding motifs (Appendix A), as do many of the promoters of the up-regulated groups 2 and 3 (peaked expression between 2 h to 4 h) (Appendix A). Thus, AP-1 might play a dominant role in both initiating the expression of other TFs during the early steps in this process, as well as coordinating the activity of the up-regulated DE TFs through PPIs. Further, beyond the expression of just the TFs, an examination of the enriched pathways among all genes whose expression peaked at 0 h (as the up-regulated group 1) by the KEGG analysis revealed an enrichment of the MAPK, NF-kappa B and TNF signaling pathways (Appendix A), in which AP-1 has been found to play a role in many immune cells [52]. Thus, AP-1 TFs may indeed play a major role in this entire process, even though itself is only transiently expressed. We speculate that this influence of AP-1 TFs, in particular, may in fact be part of the reason that activation of monocytes is needed for subsequent differentiation. We also note that our work also implicates CEBPB, only previously known to regulate the differentiation of macrophages, as also playing an important role in the early stages following the NHS addition. Thus, these TFs may prove to be potential targets for therapy during the pathological responses involving this monocyte trans-differentiation [1,5].

## 4. Materials and Methods

### 4.1. Human Primary Monocyte Isolation and the M0 Macrophage Differentiation

The informed consent from the (apparently) healthy donors was obtained for all blood samples from which the primary monocytes were derived. These donors (both male and female, between the ages of 25 to 35) were of Chinese descent. All procedures were approved by the ethics committee of Shanghai Jiao Tong University (B2020044I) at 19 October 2020. In brief, the peripheral blood mononuclear cells were isolated from the whole blood by the density gradient centrifugation using Ficoll-Paque Plus (GE Life, Bethesda, MD, USA), and the CD14+ cells were isolated from the PBMC population, according to the instructions of EasySep™ Human CD14 Positive Selection Kit II (STEM CELL technologies, Vancouver, BC, Canada). To induce the differentiation to the M0 macrophages, the CD14+ monocytes were added to the serum-free RPMI1640 medium (Thermo Fisher, Waltham, MA, USA) supplemented with 1% penicillin/streptomycin (pen/strep) (Gibco, Carlsbad, CA, USA) in T25 flasks (Corning, Steuben County, NY, USA) for 1.5 h in a cell culture incubator with 5% CO_2_ at 37°C to allow the cell adhesion and activation. Adherent cells were then cultured for seven days in the RPMI1640 medium containing 10% heat-inactivated NHS (Gemni, West Sacramento, CA, USA) and 1% Pen/Strep. Half of the culture medium was replaced by fresh medium at day four. The cells were harvested at various time points during the differentiation for the RNA-seq.

### 4.2. Flow Cytometry

The purity of isolated monocytes was detected by flow cytometric analysis (FACS). The positive group was stained with 4 μg/mL FITC mouse anti-human CD14 (BD Biosciences, San Jose, CA, USA), and the control group was stained with 4 μg/mL FITC mouse IgG2a, κ isotype control (BD Biosciences). The samples were analyzed using a BD LSRII flow cytometer (BD Biosciences). The positive gate was set, according to the staining with the isotype control. The FACS data were analyzed by the FlowJo-V10 software.

### 4.3. Immunofluorescence Microscopy

About 1 × 10^5^ cells were cultured per imaging disk in the incubator. For the immunofluorescence, cells were first fixed with 4% paraformaldehyde (PFA) at room temperature for 10 min, followed by quenching of the PFA with 125 mM glycine for 20 min. The cells were then permeabilized by incubation in 0.5% Triton X-100 for 10 min, and then incubated with 5% bovine serum albumin (BSA, Sigma-Aldrich, St. Louis, MO, USA) overnight to reduce the nonspecific binding. For the lamin B1 staining, the cells were incubated with the primary antibody rabbit anti-mouse lamin B1 (1:100 dilution by 1% BSA, Abcam, Cambridge, UK) overnight, washed three times with 1 × PBST (0.25% Tween-20 in PBS) and then incubated with the secondary antibody Alexa Fluor 647 donkey anti-rabbit IgG (1:200 dilution by 1% BSA, Abcam) for 1 h at room temperature. Finally, the nuclei were stained with DAPI (1 μg/mL). The imaging was performed with confocal fluorescence microscopy (Confocal Microscope A1, Nikon, Tokyo, Japan). For the CD14/CD68 double staining, the cells were incubated with the antibody FITC mouse anti-human CD14 (1 μg/mL, BD Biosciences) and CD68 KP1-PE (1 μg/mL, Santa Cruz, Dallas, USA) for 1.5 h at room temperature. The nuclei were stained with DAPI (1 μg/mL). The imaging was performed with fluorescence microscopy (Nikon-Eclipse-Ti, Tokyo, Japan). The images were processed using the Fiji (v1.53c, National Institutes of Health, Bethesda, MD, USA) and Imaris software (v8.1, Oxford Instruments plc, Abingdon, UK).

### 4.4. RNA-Seq Library Preparation and Sequencing

The RNA-seq was performed on several biological replicates at 11 time points (mono, 0 h, 2 h, 4 h, 8 h, 12 h, 1-day, 2-days, 3-days, 4-days and 7-days). At each time point, the total RNA from about 1 × 10^6^ cells were extracted with TRIzol (Thermo Fisher Scientific, Waltham, USA), and then treated with DNase I (final concentration at 0.2 U/μL, NEB, Ipswich, USA) to remove the genomic DNA contamination. The isolation of mRNA was performed using NEB Next PolyA mRNA Magnetic Isolation Module (NEB). The RNA-Seq libraries were prepared with KAPA Stranded mRNA-Seq Kit (KAPA Biosystems, Wilmington, NC, USA). They were sequenced on Illumina NovaSeq6000 to generate 2 × 150 bp paired-end sequencing reads.

### 4.5. RNA-Seq Quality Control and Alignment to the Human Genome

The raw reads were processed with Trimmomatic (v 0.35, USADEL Lab, Jülich, Germany) [53] to remove low-quality reads, and then the short reads (length < 100 bp) were also removed. The rRNA reads were removed by SortMeRNA. FastQC (v 0.11.5,Babraham Bioinformatics, Cambridge, UK). The clean reads were mapped to the human genome (GRCh37) with the HISAT2 (v 2.0.5, Kim Lab, Dallas, TX, USA) [54]. The gene expression levels were calculated, based on the transcripts per million (TPM) with StringTie (v 1.3.3, Johns Hopkins University, Baltimore, MD, USA) [55]. The correlation coefficients between the biological duplicates were calculated using the Spearman Rank Correlation.

### 4.6. Differentially Expressed Genes (DEGs) and the PCA Analysis

DESeq2 [56] was used to identify the differentially expressed genes between macrophages and the purified monocytes. The generalized linear models are fit for each gene, with fold-changes estimated using the empirical Bayes shrinkage. The *p* values are determined using the WALD test, with the adjustment following the procedure of Benjamini and Hochberg. The DEGs were selected with *p*. adjust < 0.01 and |log2 fold change| > 2. For this analysis, we combined the four-day sample and seven-day sample together to produce our macrophage dataset, owing to the very high correlation between these data (Spearman R > 0.98). A subset of the DEGs with high expression levels (TPM > 10 in at least one sample of monocytes or macrophages) and non-housekeeping genes was also characterized. The principal component analysis (PCA) was conducted by FactoMineR, based on the TPM of the genes. To identify the enriched annotated biological attributes in these DEGs, the gene ontology (GO) functional enrichment analysis was performed by Metascape [57]. The Kyoto Encyclopedia of Genes and Genomes (KEGG) pathway enrichment analysis was performed by clusterProfiler (v 4.2.0, Southern Medical University, Guangzhou, China), with the KEGG terms showing *p.* adjust < 0.01 and q < 0.05 considered as significantly enriched [58].

### 4.7. Identifying and Clustering the Transiently Differentially Expressed Transcription Factors

We identified the transcription factors (TFs) in our DEGs, based on the “HumanTFs” database of the annotated TFs [33]. To ensure that only those TFs with a sufficiently high expression level are considered in this analysis, we only included those with a TPM > 10. The TFs that were up-regulated or down-regulated significantly (|log_2_fold change| > 1.5, *p*. adjust < 0.01) during the monocytes activation and trans-differentiation (at 0 h, 2 h, 4 h, 8 h, 12 h, 1-day, 2-days, 3-days, 4-days and 7-days samples), compared to the purified monocytes, were regarded as up-regulated and the down-regulated differentially expressed TFs (DE TFs), respectively. We note that, with this definition, some TFs (32) were classified as both up-regulated and down-regulated, and are therefore present in both thee up-regulated and down-regulated DE TF groups. Pheatmap was used to conduct the hierarchical clustering analysis of the DE TFs. The DE TFs whose maximum difference was at an intermediate time point were classified as transient DE TFs.

### 4.8. Protein-Protein Interaction Analysis

STRING (v 11.5, European Molecular Biology Laboratory, Heidelberg, Germany) was used to construct the functional association networks of the TFs (within individual group), based on protein-protein interactions (PPIs) [59]. Only the experimental validated PPIs of the TFs were analyzed. The PPI network was visualized using Cytoscape (v. 3.9.0, Institute for Systems Biology, Seattle, WA, USA) software [60]. We identified the highly commented genes (those with more than four connections), using the cytoHubba plug-in in Cytoscape.

### 4.9. Transcription Factor (TF) Binding Site Motif Analysis

To identify the TF motifs that were enriched in the promoter regions (defined as 350 bp upstream to 50 bp downstream of the annotated transcription start site), the selected DE TF groups were examined using the HOMER software (v4.11, Benner Lab, California, USA) [25]. Only the known motif enrichment results were taken into account. We also identified the potential target genes of the selected TFs by searching for the TF binding sites within the promoter regions of other DE TFs using HOMER. For the identification of the possible target genes of AP-1, all highly expressed AP-1 members (TPM > 10 in at least one sample of mono, 0 h, 2 h or 4 h) with the available motif information in the HOMER database, were taken into account.

### 4.10. Quantitative Reverse Transcription Polymerase Chain Reaction (qRT-PCR) Analysis

The RNA used for qRT-PCR was the same total RNA as used in the aforementioned RNA-seq experiment, using DNase I (final concentration at 0.2 U/μL, NEB) to remove the genomic DNA contamination. About 40 ng RNA was reverse transcribed to the first-strand cDNA with the N6 random primer (Sangon Biotech, Shanghai, China). The qRT-PCR was performed using QuantStudio 3 Real-Time PCR system (Applied Biosystems, Carlsbad, CA, USA). Each 10-μL reaction contained 2 × Luna Universal qPCR Master Mix (NEB) and 0.5 μM of each primer (Appendix A), and all actions were performed in technical triplicate. The relative RNA expression levels were calculated, based on the 2^−dCt^ method, normalizing to the value of the actin β-subunit (ACTB), and then divided by the maximal value of each gene.

## 5. Conclusions

The transformation of human monocytes to other immune cells plays a fundamental role in the innate immunity, but our understanding of the transitional processes that underlies these transitions remains poor. Our work thus provides a foundation for future studies aiming to resolve the detailed molecular mechanisms that underlie the transition to macrophages. Of particular note, many of the transiently differentially expressed TFs were not previously implicated in this trans-differentiation, or in any immunological process. Thus, we anticipate that the future characterization of their role in the monocyte trans-differentiation to macrophages may reveal their involvement in other immune reactions. It may be that, more generally, the differentiation process between other immune cells will prove to be as complex as that described here, which can also be investigated following similar procedures as those detailed herein. It is only with the acquisition of such information that we fully understand the general principles underlying the trans-differentiation in immunity, which we expect will inform both basic biology and clinical practice.

## Figures and Tables

**Figure 1 ijms-23-15830-f001:**
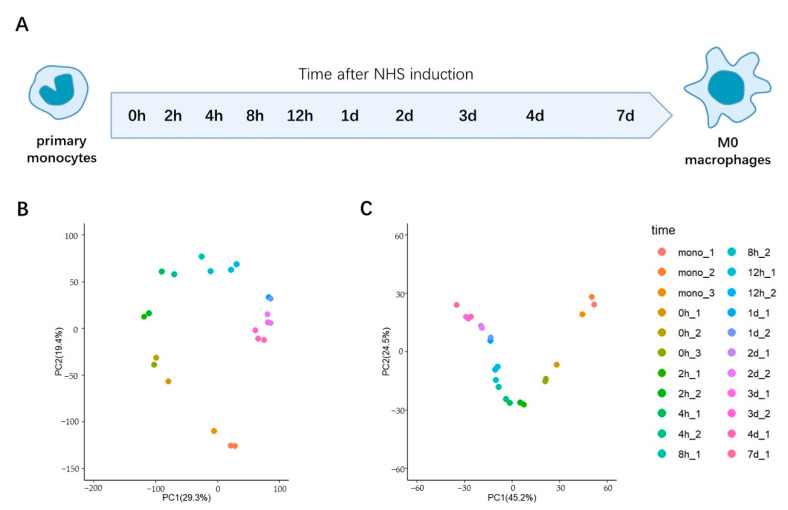
Temporal changes in the transcriptome during the trans-differentiation of monocytes to M0 macrophages. (**A**) Schematic diagram showing the examined time points during the transition. (**B**) PCA analysis of the whole transcriptome at the measured time points. (**C**) PCA analysis of the select, highly-expressed DEGs at the measured time points.

**Figure 2 ijms-23-15830-f002:**
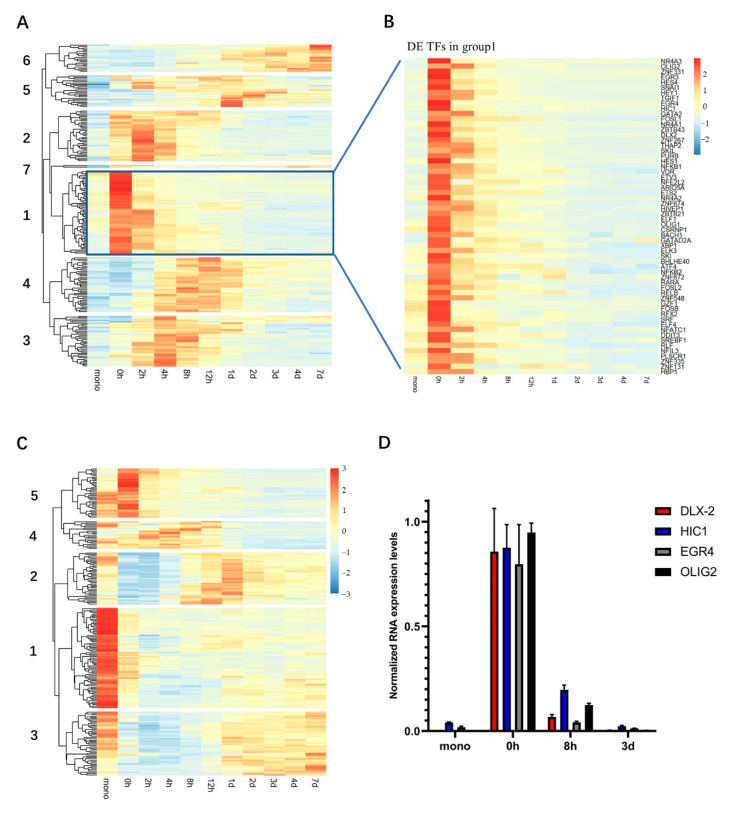
Hierarchical clustering reveals groups of the DE TFs with similar expression dynamics. (**A**) There are seven groups of up-regulated DE TFs. (**B**) Expanded view of the group 1 up-regulated DE TFs, ranked by log2(FC) (TPM). (**C**) There are five groups of the down-regulated DE TFs. (**D**) Validation of the temporal differences from the select DE TFs using qRT-PCR. The relative RNA expression levels are based on the expression of actin, and then normalized to their maximally observed level.

**Figure 3 ijms-23-15830-f003:**
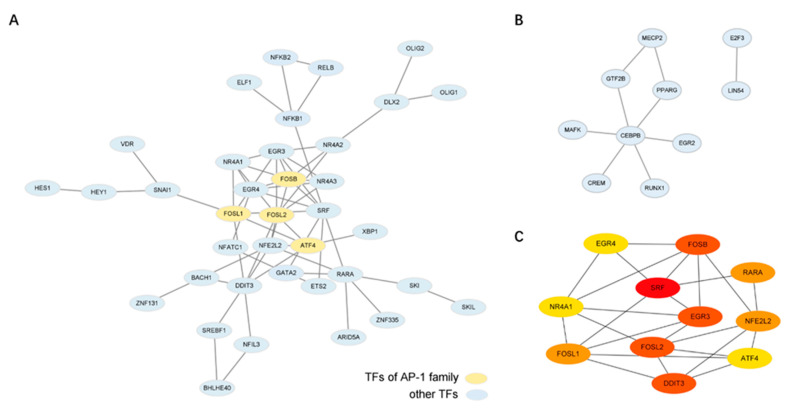
TFs that exhibit the enriched PPIs with other TFs in their group. (**A**) Up-regulated group 1 exhibits extensive PPIs. Colored in yellow are those TFs that are members of the AP-1 family (FOSB, FOSL1, FOSL2, and ATF4). (**B**) Up-regulated group 2 also exhibits a PPI network, primarily involving CEBPB. (**C**) A zoomed-in view of those genes of the up-regulated group 1 that exhibit the greatest number of connections.

## Data Availability

The RNA sequencing data produced in this study can be accessed on the NCBI Gene Expression Omnibus (GEO) under accession number GSE206082.

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
