# Peer review of "Temporal Analysis Reveals the Transient Differential Expression of Transcription Factors That Underlie the Trans-Differentiation of Human Monocytes to Macrophages"

_ijms, 2022, doi:10.3390/ijms232415830_

Round 1

Reviewer 1 Report

This paper presents data on the transcriptome analysis of monocytes during their differentiation into macrophages. Monocytes were collected from patients, transferred to artificial culturing conditions, and differentiated. The authors detailed the timing of cell transcriptome changes and obtained a large amount of sequencing data. The authors performed rather limited experimental work and focused on bioinformatic analysis of the data obtained. The authors proposed a model of sequential changes in the transcriptome.

There are the following comments.

1) It is necessary to specify how many patients were taken to obtain monocytes in each biological replica. What criteria for a "healthy" patient were used? Did the sample have specific features (residence in a certain area, belonging to a certain ethnicity)?

2) The authors need to provide convincing evidence that the most significant changes in the transcriptome (on the first day) are due to differentiation and not to the transfer of cells to artificial incubation conditions. For example, transcriptome analysis can be performed on cells transferred to culture but not exposed to a differentiation signal.

3) A more complete characterization of the original cells and derived macrophages should be provided. The original cells can be characterized by immunostaining for other markers. Do the macrophages obtained in the way described have all the typical features (increased size, increased number of lysosomal enzymes, ability to phagocytosis, synthesis of specific biologically active substances, etc.)?

4) In addition, additional characteristics of cells at intermediate stages are needed. The cells after 1 day of incubation are of particular interest.

5) Comments on Figure S2 - cells differ markedly in size, although the scale bar is the same size. Negative controls are not presented for immunostaining. Images of the nucleus (lamin staining) are given only for 2-3 cells in unfavorable projections.

These remarks will increase the biological significance of the obtained results and give their strict and correct interpretation.

Reviewer 2 Report

The manuscript entitled " Temporal analysis reveals transient differential expression of transcription factors underlying trans-differentiation of human monocyte to macrophage” illustrates the requirement for dynamic expression of transcription factors during trans-differentiation. The manuscript is well written. However, it would be strengthened by addressing the following points:

1.       In the methods section please provide more detail. Are the blood samples all from the same person for each experiment or a mixed population. What are the sex and age range of the individuals the samples were collected from.

2.       Please indicate in the method section as to how the statistical analysis was done.

3.       The manuscript does not provide any direct evidence that any of the noted differences in TF’s are involved in this differentiation process. Has this been shown before. This needs to be addressed in some manner in the manuscript.

Reviewer 3 Report

The research article prepared by W. Deng et al. entitled ‘Temporal analysis reveals transient differential expression of transcription factors underlying trans-differentiation of human monocyte to macrophage’ aims to analyse the temporal changes of the transcriptome during trans-differentiation of primary human monocytes into M0 macrophages in vitro.The study is well structured, however the article is not very well organized and needs editing in order to make its intelligible.

Comments and Suggestions for Authors:

-Introduction is too long. The description of the results should be excluded

-Correction of gens nomenclature is needed, italics, please (line 77)

-Check and correct the type and size of the font, please (line 341,351)

-There is no need to describe methods in the Results section. This generates repetitions in the manuscript (line 126-136).

-Please provide Discussion in a separate section and discuss your results with the other authors’ findings

Round 2

Reviewer 1 Report

The authors conducted a critical analysis of the experiments performed, revised the figures, and presented a more accurate and correct interpretation of the data obtained. A detailed response to the comments is presented.

In the corrected form the article can be recommended for publication in the journal.

Reviewer 3 Report

The authors improved the manuscript according to my suggestions. I recommend the paper for publication.